# An approach for projecting the timing of abrupt winter Arctic sea ice loss

Camille Hankel[1] and Eli Tziperman[1,2]

[1]Department of Earth and Planetary Sciences, Harvard University, 20 Oxford St, Cambridge, MA 02138
[2]School of Engineering and Applied Sciences, Harvard University

**Correspondence:** Camille Hankel (camille_hankel@g.harvard.edu)

**Abstract.** Abrupt and irreversible winter Arctic sea-ice loss may occur under anthropogenic warming due to the disappearance
of a sea-ice equilibrium at a threshold value of $CO_2$, commonly referred to as a tipping point. Previous work has been unable
to conclusively identify whether a tipping point in winter Arctic sea ice exists because fully-coupled climate models are
too computationally expensive to run to equilibrium for many $CO_2$ values. Here, we explore the deviation of sea ice from
its equilibrium state under realistic rates of $CO_2$ increase to demonstrate for the first time how a few time-dependent $CO_2$
experiments can be used to predict the existence and timing of sea-ice tipping points without running the model to steady-state.
This study highlights the inefficacy of using a single experiment with slow-changing $CO_2$ to discover changes in the sea-ice
steady-state, and provides a novel alternate method that can be developed for the identification of tipping points in realistic
climate models.

## 1   Introduction

The Arctic is warming at a rate at least twice as fast as the global mean with profound consequences for its sea ice cover.
Sea ice is already exhibiting rapid retreat with warming, especially in the summertime, (Comiso and Parkinson, 2004; Nghiem
et al., 2007; Stroeve et al., 2008; Notz and Stroeve, 2016; Stroeve and Notz, 2018), shortening the time that socioeconomic and
ecological systems have to adapt. These concerns have motivated a large body of work dedicated to both observing present-day
sea ice loss (Kwok and Untersteiner, 2011; Stroeve et al., 2012; Lindsay and Schweiger, 2015; Lavergne et al., 2019) and
modeling sea ice to understand whether its projected loss is modulated by a threshold-like or "tipping point" behavior. Abrupt
loss of Arctic sea ice could be driven by local positive feedback mechanisms (Curry et al., 1995; Abbot and Tziperman, 2008;
Abbot et al., 2009; Kay et al., 2012; Leibowicz et al., 2012; Burt et al., 2016; Feldl et al., 2020; Hankel and Tziperman, 2021),
remote feedback mechanisms that increase heat flux from the mid-latitudes (Holland et al., 2006; Park et al., 2015), or by
the natural threshold corresponding to the seawater freezing point (Bathiany et al., 2016). If such an abrupt loss is caused by
irreversible processes (typically, strong positive feedback mechanisms as opposed to the reversible mechanism of a freezing
point threshold of Bathiany et al., 2016), it is referred to here as a "tipping point". A tipping point in the sense used here is
a change in the number or stability of steady-state solutions (Ghil and Childress, 1987; Strogatz, 1994) as a function of $CO_2$
and is also known as a bifurcation. We note that some of the climate literature uses "tipping points" in a more general sense

of a relatively rapid change (e.g., Lenton, 2012). While most studies have concluded that there is no tipping point during the transition from perennial to seasonal ice cover (i.e., during the loss of *summer* sea ice), the existence of a tipping point during the loss of *winter* sea ice (transition to year-round ice-free conditions) continues to be debated in the literature (Eisenman, 2007; Eisenman and Wettlaufer, 2009; Notz, 2009; Eisenman, 2012). Wagner and Eisenman (2015) showed that a winter tipping point disappeared from a simple model of sea ice with no active atmosphere when a longitudinal dimension was added. On the other hand, other literature (e.g., Abbot and Tziperman, 2008; Hankel and Tziperman, 2021) has demonstrated the importance of atmospheric feedbacks, not included in the model of Wagner and Eisenman (2015), in inducing winter sea ice tipping point. Furthermore, three out of seven fully-complex Global Climate Models (GCMs) that lost their winter sea ice completely in the CMIP5 Extended RCP8.5 Scenario showed a very abrupt change in winter Arctic sea ice resembling a tipping point (Hezel et al., 2014; Hankel and Tziperman, 2021). However, given the projected rapid changes to $CO_2$ in the coming centuries and the slower response of the climate system, we do not expect future sea ice to be fully equilibrated to the $CO_2$ forcing at a given time, making the standard steady-state tipping point analysis challenging. Thus, our first goal is to understand abrupt winter Arctic sea ice changes—which may or may not be due to tipping points—under rapidly changing $CO_2$ forcing, where sea ice is not at equilibrium.

Tipping points imply a bi-stability (meaning that sea ice can take on different values for the same $CO_2$ concentration), and hysteresis — an irreversible loss of sea ice even if $CO_2$ is later reduced. Bi-stability (and therefore tipping points) can be tested for by running model simulations to steady-state at many different $CO_2$ values, which is computationally inefficient in expensive, state-of-the-art GCMs. GCM studies, therefore, tend to use a single experiment with very gradual $CO_2$ increases and decreases (Li et al., 2013) or even a faster $CO_2$ change (Ridley et al., 2012; Armour et al., 2011) and look for hysteresis in sea ice that would imply the existence of a tipping point. These studies implicitly assume that such a run should approximate the behavior of the steady-state at different $CO_2$ concentrations. However, Li et al. (2013) further integrated two apparently bi-stable points and found that they equilibrated to the same value of winter sea ice: there was no "true" bi-stability at these two $CO_2$ concentrations, the sea ice was simply out of equilibrium with the $CO_2$ forcing. This calls into question the current use of time-changing $CO_2$ runs to study the bifurcation structure of sea ice.

In light of the difficulties in using climate model runs with time-changing $CO_2$ (hereafter "transient runs"), the first goal of this work is to understand the relationship between these transient runs and the steady-state value of sea ice in systems with and without bifurcations (since the existence of a bifurcation in winter sea ice remains unknown), and the second goal is to develop a new efficient method for the identification of tipping points from transient runs. Theoretical work in dynamical systems (Haberman, 1979; Mandel and Erneux, 1987; Baer et al., 1989; Tredicce et al., 2004) and studies related to bi-stability in the Atlantic Meridional Overturning Circulation (Kim et al., 2021; An et al., 2021) have examined systems with tipping points when the forcing parameter ($CO_2$ in our case) changes in time at a finite rate. They found that as the forcing parameter passes the bifurcation point, the system continues to follow the old equilibrium solution for some time before it rapidly transitions to the new one. Specifically, (Kim et al., 2021; An et al., 2021) find that the width of the hysteresis loop of AMOC is altered by the rate of forcing changes– this phenomenon is referred to as "rate-dependent hysteresis". This rate-dependence occurs in their case in a system that also has bi-stability and hysteresis in the equilibrium state. This type of analysis has, to our knowledge,

not yet been applied in the context of winter sea ice loss under time-changing $CO_2$ concentrations, nor compared in systems with and without a bifurcation (that is, with and without an equilibrium hysteresis).

In order to analyze how the hysteresis of sea ice under time-changing forcing relates to the steady-state behavior of sea ice, we run a simple physics-based model of sea ice (Eisenman, 2007), configured in three different scenarios: with a large $CO_2$ range of bi-stability, a small range of bi-stability, and no bi-stability in the equilibrium. These three scenarios span the range of possible behaviors of winter sea ice in state-of-the-art climate models. Each case is run with different rates of $CO_2$ increase (ramping rates). We use results from this model and from an even simpler standard 1D dynamical system to demonstrate that the convergence of the transient behavior (under time-changing forcing) to the equilibrium behavior is very slow as a function of the ramping rate of $CO_2$. In other words, even climate model runs with very slow-changing $CO_2$ forcing may simulate sea ice that is considerably out of equilibrium near the period of abrupt sea ice loss. Finally, we propose a novel approach for uncovering the underlying equilibrium behavior—and thus the existence and location of tipping points—in comprehensive models where it is computationally infeasible to simulate steady-state conditions for many $CO_2$ values. Such a method is important given the model-dependent nature of winter sea ice tipping points discussed above; uncovering the existence of sea ice tipping points in GCMs, which are the most realistic representation of Arctic-wide sea ice behavior that we have, is the next step toward understanding whether such tipping points exist in the real climate system. Our goal has some parallels to that of Gregory et al. (2004), who used un-equilibrated GCM runs to deduce the equilibrium climate sensitivity when fully-equilibrated runs were computationally infeasible.

As mentioned above, some GCMs exhibit an abrupt change in winter sea ice that may be a tipping point, and others do not (Hezel et al., 2014; Hankel and Tziperman, 2021). The reasons likely involve numerous differences in parameters and parameterizations. It is not obvious how to modify parameters in a single GCM to display all of these different behaviors. Therefore, we choose to use an idealized model of sea ice where we can directly produce different bifurcation behaviors to address our second goal and answer the question: is it possible to identify the $CO_2$ at which tipping points occur without running the model to a steady state for many $CO_2$ values? Answering such a question in a simple model is an obvious prerequisite to tackling the problem of identifying climate bi-stability in noisy, high-dimensional, GCMs. In order to perform this analysis for each of the three scenarios mentioned above, we modify the strength of the albedo feedback via the choice of surface albedo parameters. The albedo values used here to generate the three scenarios are not meant to reflect realistic albedo values, but rather allow us to represent in a single model the range of sea ice equilibria behaviors that may exist in different GCMs. We, therefore, follow in the footsteps of previous studies (e.g., Eisenman, 2007) that have also changed parameters outside of their physically relevant regime in order to understand *summer* sea ice bifurcation behavior; here we follow the same approach to understand when a *winter* sea ice bifurcation can be detected without running an expensive climate model to steady-state.

## 2 Methods

### 2.1 Sea ice model

The sea ice model used follows Eisenman (2007) almost exactly and its key features are depicted schematically in Figure 1. The model contains four state variables: sea ice effective thickness ($V$, which is volume divided by the area of the model grid box), sea ice area ($A$), sea ice surface temperature ($T_i$), and mixed layer temperature ($T_{ml}$) for a single box representing the entire Arctic. Subsequent versions of this sea ice model have been used in Eisenman and Wettlaufer (2009), Eisenman (2012), and Wagner and Eisenman (2015). Those versions are derived from the model used here, making a few further modest simplifications (using a hyperbolic tangent function for surface albedo, assuming the ice surface temperature is in a steady state, combining all prognostic variables into one, enthalpy) that do not affect the qualitative behavior of the model (i.e., the nature of summer and winter sea ice bifurcations). We choose to implement the earlier model because it explicitly represents the key physical variables of ice volume, area, ocean temperature, and ice temperature as prognostic variables — as opposed to combining them all into a single enthalpy — and thus provides more transparency and interpretability. We, therefore, do not expect our results to change if we use any of the later model versions.

In the model, the atmosphere is assumed to be in radiative equilibrium with the surface, and the model is forced with a seasonal cycle of insolation, of poleward atmospheric heat transport from the mid-latitudes, and of local optical thickness of the atmosphere, which represents cloudiness. Sea ice growth and loss are primarily determined by the heat budget at the bottom of the ice and are therefore set by the balance between ocean-ice heat exchanges, and heat loss through the ice to the atmosphere. When conditions for surface melting are met (when the ice surface temperature is zero and net fluxes on the ice are positive), all surface heating goes into melting ice and the surface albedo of the ice is set to the melt pond albedo. The ocean temperature is affected by shortwave and longwave fluxes in the fraction of the box that is ice-free, and by ice-ocean heat exchanges. When the ocean temperature reaches zero, all additional cooling goes into ice production while the ocean temperature remains constant. The full equations of the sea ice model can be found in the original paper (Eisenman, 2007) and in the online Supporting Information; here, we highlight a few minor ways in which our implementation differs. First, for simplicity, we do not model leads, which in the original model were represented by capping the ice fraction at 0.95 rather than 1. Second, we use an approximation to the seasonal cycle of insolation (Hartmann, 2015) using a latitude of 75N. The atmospheric albedo is set to 0.425 to produce the same magnitude of the seasonal cycle as in the original model of Eisenman (2007).

### 2.2 Setup of simulations

In our transient-forcing scenarios (described below), we vary $CO_2$ in time which affects the prescribed near-surface atmospheric mid-latitude temperature ($T_{\mathrm{mid-lat}}$) and the atmospheric optical depth ($N$, see Supporting Information). Specifically, we increase the annual mean of $T_{\mathrm{mid-lat}}$ by 3 °C per $CO_2$ doubling and $N$ by a $\Delta N$ that corresponds to 3.7 W/m$^2$ per doubling. All model parameters are as in Eisenman (2007) except as mentioned below.

122  We configure the model in three different scenarios that yield a wide $CO_2$ range of bi-stability in winter sea ice (Scenario
123 1), a small range of bi-stability in winter sea ice (Scenario 2), and no bi-stability in winter sea ice (Scenario 3). We do so by
124 modifying the strength of the ice-albedo feedback by changing the albedos of bare ice ($\alpha_i$), melt ponds ($\alpha_{mp}$), and ocean ($\alpha_o$),
125 as listed in Table S1.

126  In each of the three scenarios, we tune the model (by adjusting the mean and amplitude of the atmospheric optical depth)
127 to roughly match the observed seasonal cycle of ice thickness under pre-industrial $CO_2$ ($\sim$ 2.5–3.7 m, Eisenman, 2007). We
128 then run each scenario with multiple $CO_2$ ramping rates (expressed in "years per doubling") with an initial stabilization period
129 (fixed pre-industrial $CO_2$), a period of exponentially increasing $CO_2$ concentration (which corresponds to linearly increasing
130 radiative forcing), another period of stabilization at the maximum $CO_2$, a period of decreasing $CO_2$, and a final period of
131 stabilization at the minimum $CO_2$ value (see Supplemental Figure S2). Scenarios 2 and 3 are ramped to higher final $CO_2$
132 values than Scenario 1 so that they lose all their sea ice. We also directly calculate the steady-state behavior of the sea ice (as
133 done in the original study) by running many simulations with fixed $CO_2$ values until the seasonal cycle of all the variables
134 stabilizes. Because we expect multiple equilibria (which could be ice-free, seasonal ice, or perennial ice) at some $CO_2$ values
135 in Scenarios 1 and 2, we run these steady-state simulations starting with both a cold (ice-covered) and a warm (ice-free) initial
136 condition in order to find these different steady-states. In the ice-free initial condition runs, the ice-albedo feedback will still
137 play an important role if the temperature cools sufficiently for ice to develop. At $CO_2$ values for which the sea ice is bi-stable,
138 the ice-free initial condition evolves to a perennially ice-free steady-state, and the ice-covered initial condition evolves to a
139 seasonally ice-covered steady-state (seen by the dotted and dashed lines respectively in Figs. 2a and 2c).

140 **2.3 Cubic ODE**

141 The main points we are trying to make about the transient versus equilibrium behavior of winter sea ice near a tipping point
142 are not unique to the problem of winter sea ice, and in order to demonstrate this, we use the simplest mathematical model
143 that can display tipping points, following other studies that have also used such simple dynamical systems (Ditlevsen and
144 Johnsen, 2010; Bathiany et al., 2018; Ritchie et al., 2021; Boers, 2021). The cubic ODE used, while much simpler than the
145 sea ice model above, has some of the key characteristics of the sea ice system (it is a non-autonomous system due to the time-
146 depending forcing and has saddle-node bifurcations), which allows for direct comparison between the two models. The ODE
147 equation,

148 $$\frac{dx}{dt} = -x^3 + \delta x + \beta(t), \qquad \beta(t) = \beta_0 + \mu t, \tag{1}$$

149 contains a time-changing forcing parameter, $\beta(t)$ mimicking the effects of $CO_2$ in the sea ice model. We consider this differ-
150 ential equation in three scenarios, paralleling those used with the sea ice model: in Scenario 1, $\delta = 5$ leading to a wide region
151 of bi-stability; in Scenario 2, $\delta = 1$ leading to a narrow region of bi-stability, and finally, in Scenario 3, $\delta = 0$ leading to a
152 mono-stable system. The different values of $\delta$, therefore, produce the same three scenarios that were achieved in the sea ice
153 model by modifying the strength of the ice-albedo feedback. We mimic the hysteresis experiments of the sea ice model with
154 a sequence of ramping up and ramping down (using different ramping rates, $\mu$) with values of $\beta$ ranging from $-10$ to $10$ to

sweep the parameter space that contains the bifurcations. We calculate the steady-states with fixed values of $\beta$ ($\mu = 0$), starting
with both a positive and a negative initial condition of $x$ to yield two stable solutions when these exist.

157       We want to calculate the upper and lower $CO_2$ values of the hysteresis region in runs with time-changing (i.e., transient)

$CO_2$ forcing. We do so by calculating the $CO_2$ value at which the March sea ice area drops below a critical threshold (50%
ice coverage; results are insensitive to the specific value used) during increasing and decreasing $CO_2$ integrations: we denote
these $CO_2$ values $CO_2^i$ and $CO_2^d$, respectively (see Supplemental Figure S9). The difference between $CO_2^i$ and $CO_2^d$ is referred
to below as the "hysteresis width" of the rate-dependent hysteresis whether an equilibrium hysteresis exists or not; this width
approaches the width of bi-stability at very slow ramping rates.

## 2.4   A new method for predicting the $CO_2$ of the sea ice tipping point

One of our main goals (see Introduction) is to efficiently estimate the equilibrium behavior of sea ice, including the location of
tipping points, without running the model to a steady state for many $CO_2$ values. This would show that such estimation could
be calculated for GCMs where tipping points cannot be detected using steady-state runs due to their computational cost. In
order to estimate the values of $CO_2^i$ and $CO_2^d$ that would have occurred for an infinitely slow ramping rate (in other words, the
range of $CO_2$ for which there is bi-stability) using only the transient runs, we fit a polynomial of the form $f(x) = mx^c + b$ to
$CO_2^i$ and $CO_2^d$ as functions of the ramping rate $x$. Because $c$ is negative, the fitted parameter $b$ represents the prediction of $CO_2^i$
and $CO_2^d$ at infinitely slow ramping rates, i.e., in the steady state. We also calculate the uncertainty on the fitted parameter $b$ by
block-bootstrapping to account for auto-correlation; see Supporting Information. Other fits to $CO_2^i$ and $CO_2^d$ as a function of
ramping rates, such as an exponential function $f(x) = a + b\exp(-cx)$ could in principle be used, although we found that fit to
be less good in our case.

## 3   Results

In the following three subsections, we discuss the behavior of the sea ice model and the cubic ODE under time-changing
forcing, the relationship of the transient and equilibrium behaviors, and a method that we propose for inferring the existence
and location of tipping points from the transient behavior. Equilibrium hysteresis refers here to the path-dependent solution of
a variable due to bi-stability and a bifurcation in the steady-state (in other words, the loop traced by the steady-state solutions).
The term "rate-dependent hysteresis" (An et al., 2021; Manoli et al., 2020) describes hysteresis loops that appear in time-
changing forcing runs (rather than in the steady state) and that depend on the rate of forcing change. In our analysis "rate-
dependent hysteresis" applies to both systems with and without equilibrium hysteresis: it refers to any differences in the results
for increasing vs. decreasing $CO_2$ simulations of sea ice that are altered by the rate of $CO_2$ change.

### 3.1   Transient response of Arctic winter sea ice to time-changing $CO_2$

Our goal in this section is to understand the relationship of winter sea ice forced with time-changing $CO_2$ to its equilibrium
state, both in cases with and without a sea ice tipping point. In Figs. 2a,c,e, we plot the results of running all three scenarios
(wide range of bi-stability (Scenario 1), narrow range of bi-stability (2), and no bi-stability (3)) under time-changing (transient)
and fixed $CO_2$ values. In all scenarios, the experiments run with time-changing $CO_2$ exhibit rate-dependent hysteresis; the
hysteresis width (lower horizontal gray bar in Fig. 2a) is larger for faster ramping rates (Figs. 2a,c,e). For Scenarios 1 and
2, which have a region of bi-stability and equilibrium hysteresis (upper gray bar in Fig. 2a), this corresponds to a widening
from the equilibrium hysteresis (that would exist even with infinitely slow ramping rates), while in Scenario 3, this hysteresis
occurs only in transient simulations and is due to the inertia in the system (the sea ice can't respond instantaneously to forcing
changes). In Scenarios 1 and 2, whose equilibrium solutions (dashed and dotted black lines in Fig. 2) have a tipping point
and therefore an infinite gradient of sea ice thickness vs. $CO_2$, the faster ramping rates also lead to more gradual (and finite)
gradient of sea ice thickness vs. $CO_2$.

195        The rate-dependent hysteresis loops across all scenarios at fast enough ramping rates (loops composed of the darkest blue

and darkest red) are qualitatively similar in shape, despite their different underlying steady-state structures. This similarity
indicates that from a single hysteresis run with time-changing $CO_2$ we cannot discern whether the underlying Arctic winter
sea ice equilibrium behavior has a region of bi-stability or not, nor how wide the region of true bi-stability is. In particular, a
single hysteresis loop found from a time-changing forcing simulation would always overestimate the width of bi-stability if it
was assumed to represent a quasi-steady state. This result demonstrates that the apparent sea ice hysteresis loop found by Li
et al. (2013) could be due to a system without an equilibrium hysteresis, as they suggest, or due to a system with a narrower
equilibrium hysteresis than the one implied by their transient simulation.

203        We now discuss the behavior of the simple cubic ODE (Eqn. 1) under similarly time-changing forcing. Previous work in

the dynamical systems literature (e.g., Haberman, 1979; Mandel and Erneux, 1987; Baer et al., 1989; Breban et al., 2003;
Tredicce et al., 2004; Kaszás et al., 2019) has examined a variety of simple systems to understand the nature of bifurcations in
the presence of a time-changing ("drifting" or "transient") forcing parameter. In the climate literature as well (e.g., Ditlevsen
and Johnsen, 2010; Bathiany et al., 2018; Ritchie et al., 2021; Boers, 2021), idealized dynamical systems similar to our Eqn. 1
have been used to understand the predictability of tipping points in the presence of noise, and the ability to recover from such
tipping points ("overshoot" scenarios). These works, as well as the AMOC study of An et al. (2021), found that a system with
a bifurcation that is run with a time-changing forcing parameter can follow a given equilibrium value beyond the bifurcation
value of the forcing parameter before undergoing the tipping point transition to the new equilibrium value. This is consistent
with the out-of-equilibrium behaviors we find for sea ice in Scenarios 1 and 2. To our knowledge, the simple ODE used here
has not yet been analyzed with our specific goal in mind: to compare the shape of rate-dependent hysteresis loops in generic
dynamical systems both with and without bifurcations, and to address the question of whether the equilibrium behavior can be
inferred from the rate-dependent behavior of such systems.

216        To address these two goals, we configure Eqn. 1 analogously to the sea ice model in three scenarios with wide bi-stability

(Scenario 1), narrow bi-stability (Scenario 2), and no bi-stability (Scenario 3) and force it with a time-changing forcing pa-
rameter. In Figs. 2b,d,f, we see that the three scenarios with similar dynamics (but different equilibrium structures) all display
rate-dependent hysteresis, similar to the result from the sea ice model. Specifically, even when there is only one stable equilib-
rium solution in both models (Scenario 3, panels e and f), there is still a narrow region of rate-dependent hysteresis. Thus, we
find that the inability to tell if rate-dependent hysteresis in Arctic winter sea ice is accompanied by an underlying equilibrium
hysteresis appears to be a generic feature of dynamical systems, which helps explain the challenges of interpreting the results
of Li et al. (2013).
Mathematically, this 1D system is fundamentally different from the sea ice model because it is not periodically forced. We
show in the Supporting Information that adding a sinusoidal forcing term to the ODE does not qualitatively change our results.

## 3.2   Slow convergence of the rate-dependent hysteresis to the equilibrium behavior

Our next objective is to demonstrate that it would require expensive runs in a GCM to approach the equilibrium behavior of
sea ice using slower and slower-changing $CO_2$ runs (hysteresis experiments). As we saw in Fig. 2, the rate of loss of sea ice
with increasing $CO_2$ is infinite (dashed and dotted black lines) in Scenarios 1 and 2 at the tipping points. On the other hand, the
gradient of sea ice thickness with respect to $CO_2$ is more gradual and finite under time-changing forcing (blue and red curves)
but steepens as the ramping rate of $CO_2$ decreases. We now quantify the rate of this steepening by examining the maximum
gradient of sea ice loss during each transient simulation as a function of ramping rate (inverse of the years per doubling of
$CO_2$).
In Fig. 3a, we plot the maximum gradient of March sea ice thickness *with respect to $CO_2$* during each hysteresis experiment,
as a function of the $CO_2$ ramping rate. In Scenarios 1 and 2 (wide and narrow bi-stability, respectively), the maximum gradient
gets greater as the ramping rate is slower (Fig. 3a, negative slopes of solid and dashed lines), consistent with Fig. 2 (e.g.,
steepening from dark blue to light blue curves in Figs. 2a,b). In particular, the gradient approximately follows a negative power
law as a function of ramping rate on both warming and cooling time series. In Scenario 3, the maximum gradient is nearly
insensitive to the ramping rate (relatively flat dash-dotted lines). In Fig. 3b, we see a similar result for the simple ODE, as seen
by the shallowing of the power law from Scenarios 1 to 3 (though here the slope in Scenario 3 is clearly nonzero). Notably,
in the cubic ODE the power law in the case with the largest region of bi-stability (Scenario 1) is approximately given by
$\max(dx/d\beta) \propto \mu^{-1}$, where $\mu$ again is the ramping rate. The Supporting Information further explains the above convergence
rate of $\mu^{-1}$.
A dependence of the maximum gradient on (ramping rate)$^{-1}$ in the case of wide bi-stability suggests that running a climate
model with twice as gradual $CO_2$ ramping leads to less than a factor of two increase in the gradient $\max(dV/dCO_2)$. This is
an important result because this implies that the distance between the $CO_2$ at the simulated transient "tipping point" and the
$CO_2$ of the true (equilibrium) tipping point (which we want to estimate) also only reduces by a factor of two when the ramping
rate is reduced by a factor of two. A greater power law slope (e.g., a slope of $-2$) would imply a much faster convergence to
the equilibrium location of the tipping point. Thus, using more and more gradual ramping experiments may be an inefficient
way to approach the equilibrium behavior of this physical system, suggesting the need for a more efficient approach, discussed
next.

## 3.3 Predicting the steady-state behavior of sea ice using only transient runs

Our main novel result, presented next, is a method for finding the $CO_2$ concentration at which a bifurcation (if any) occurs in the equilibrium using computationally feasible transient model runs instead of fixed-forcing steady-state runs. We are interested in this $CO_2$ concentration because it determines the threshold beyond which significant sea ice loss is practically irreversible (Ritchie et al., 2021). In our simple, inexpensive model, we can test the estimates of the bi-stability and associated tipping points derived from transient model runs against the known true tipping points and equilibrium structure that are found from fixed-forcing runs (see Methods). When used in a GCM, our method would provide a prediction for the existence and location of tipping points when the equilibrium value of sea ice is actually unknown. Thus, this section is a proof of concept that our new method can accurately determine whether observed rate-dependent hysteresis is caused by lag around a system with no bi-stability or tipping points or caused by a rate-dependent widening of an equilibrium hysteresis loop in a system with tipping points.

In Fig. 4a, we plot a measure of the upper and lower $CO_2$ values that correspond to the rightmost and leftmost edges of the rate-dependent hysteresis (by calculating the $CO_2$ at which the March sea ice area crosses a critical threshold, see Methods and Supplementary Figure S9). We plot this threshold for the warming (increasing greenhouse concentration) trajectories in blue ($CO_2^i$) and for the cooling (decreasing greenhouse) trajectories in red ($CO_2^d$), as a function of the ramping rate for all three scenarios. As expected, as the ramping rate gets slower $CO_2^i$ and $CO_2^d$ asymptote to the $CO_2$ values corresponding to the edges of the equilibrium hysteresis and the location of the true tipping points in the case of Scenarios 1 and 2 (denoted by the $\times$ symbols). In Scenario 3, $CO_2^i$ and $CO_2^d$ asymptote to the same value (the rate-dependent hysteresis width approaches zero) because there is no bi-stability in the steady-state.

Finally, we demonstrate that fitting a curve to the edges of the rate-dependent hysteresis ($CO_2^i$ and $CO_2^d$) as a function of the ramping rate can be used to predict $CO_2^i$ and $CO_2^d$ at infinitely slow ramping rates (i.e., the edges of the equilibrium hysteresis). This would allow us to estimate the $CO_2$ value corresponding to a bifurcation in the equilibrium behavior without running a model to a steady state. In Fig. 4a, we plot $CO_2^i$ and $CO_2^d$, and the curves that fit them (see Methods) as functions of the ramping rate, and the predicted values of $CO_2^i$ and $CO_2^d$ at infinitely slow ramping rates with a 95% confidence interval range shaded around them. We perform this fitting and estimation process using all the ramping experiments (18 different ramping rates total, as shown in Fig. 4a). We then repeat the fit using fewer and fewer experiments to explore how the uncertainty on predicted values of $CO_2^i$ and $CO_2^d$ increases as we move to only using a few fast ramping experiments that are more feasible when using full complexity climate models. Fig. 4b shows a summary of these analyses.

The predicted values of $CO_2^i$ and $CO_2^d$ are remarkably accurate for all scenarios (points approaching the red and blue $\times$ in Fig. 4b), even when excluding several of the slower ramping experiments. This is an important test because when this method is applied to a GCM, one would only have a smaller number of faster ramping experiments due to computational limitations. The uncertainties (indicated by the shaded blue and red bars around the points) in the predictions grow when excluding more experiments from the curve fitting process but still remain very low, especially for Scenarios 1 and 2. In predicting $CO_2^d$ for Scenario 3, the uncertainties are a bit higher because the functional form of our fit does not represent this case as well as

the others, leading to serial correlation in the residuals. The structure in the residuals can be used to guide the choice of the functional form used to fit such data in future applications. This same method and functional form can also successfully predict the equilibrium structure of our simple ODE (Eqn. 1), with even smaller uncertainties on the prediction when using very few ramping experiments (see Figure S11). Finally, we can use the difference of the distributions $CO_2^i$ and $CO_2^d$ to calculate the probability that bi-stability—and thus a tipping point—exists (see Supporting Information). Another very similar approach using only the difference between $CO_2^i$ and $CO_2^d$ (i.e., the hysteresis width) as a function of the ramping rate is also shown in Figure S10.

Overall, these results demonstrate the potential for using several shorter runs with time-changing $CO_2$ forcing to efficiently estimate the $CO_2$ value of the tipping points and predict the existence of bi-stability in GCMs where equilibrium runs or long, slow-ramping hysteresis runs are computationally infeasible.

## 4 Discussion

We have shown that a single climate model hysteresis run with time-changing (transient) forcing cannot be used to conclusively estimate the true location of Arctic winter sea ice tipping points, the range of bi-stability in the steady-state, and even the existence of bi-stability at all. We demonstrated that the transient sea-ice responses under a time-changing $CO_2$ reflect the generic behavior of a nonlinear dynamical system (e.g., our Eqn. 1): specifically, we showed that systems with and without bi-stability can also produce qualitatively indistinguishable rate-dependent hysteresis behavior. We also find that very long model runs are needed to identify whether the system approaches a bifurcation (Fig. 3) and at what $CO_2$ this occurs. We showed that even in runs with a very slow-changing $CO_2$, the system can be surprisingly far from the equilibrium as it undergoes a tipping point, consistent with the work of Li et al. (2013). In addition, even with a very slow ramping experiment, one would always have to perform additional expensive fixed-forcing experiments (as done by Li et al., 2013) to confirm that the experiment was indeed in quasi-equilibrium. Instead, we propose a novel method that uses a few fast-ramping experiments to efficiently predict the true range of bi-stability and provide uncertainty estimates on this prediction.

We demonstrated that the method we propose can accurately predict the steady-state behavior of sea ice in a simple model; now we discuss applying this method to a GCM. First, we note that while we use a highly idealized model of sea ice in this study, the method developed deals with identifying bi-stability in complex systems with unknown equilibrium structures more generally. This means that the framework should be applicable to other models (including GCMs), since moving from fast to slower ramping rates allows convergence to the equilibrium behavior. It could also be used in the context of vastly different climate problems, for example, in identifying the abrupt transitions to a moist greenhouse (Popp et al., 2016), runaway greenhouse (Goldblatt et al., 2013), or snowball Earth state (Hyde et al., 2000). The functional form used to fit the transient runs, as well as the level of certainty achieved from a given number of experiments, would likely depend on the given model and climate problem analyzed. Possible challenges in finding the functional best fit to the transient runs might mirror those of Gregory et al. (2004) who encountered difficulties when trying to fit a line to un-equilibrated GCM runs with a different goal

of deducing the equilibrium climate sensitivity. We suggest that a careful examination of the residuals from a given fit can help
guide the choice of functional form.
The generality of the method also highlights another advantage: the same set of ramping experiments in a GCM could be
used to analyze all suspected tipping elements in the Earth's climate system simultaneously. The main challenge we anticipate
in applying this method to GCMs comes from the significant stochastic variability and multiple timescales of forcings that
may render the calculated width of the rate-dependent hysteresis more uncertain in a GCM. Nonetheless, using multiple runs
to estimate the width of the bi-stability of a given climate variable and providing a quantified uncertainty on such a prediction
should offer a potential improvement over using a single hysteresis experiment.
We can estimate the efficiency of the proposed approach over more standard ones when applied in a GCM. Taking the
experimental setup of Li et al. (2013) as a guide, we can assume that a slow-ramping experiment to $4\times CO_2$ requires a 2000-
year ramp up and ramp down with at minimum a 2500-year equilibration period after each ramp (though they actually allowed
the model to equilibrate for nearly 6000 years). Within the 500 ppm width of the rate-dependent hysteresis found by Li et al.
(2013), ten fixed-forcing experiments 2500 years long would be needed to test for bi-stability and estimate the tipping point
location at a relatively crude accuracy of 100 ppm. This leads to a total of 34,000 simulation years. On the other hand, if we
used our proposed approach, we could run three ramping experiments with fast to intermediate rates of 100, 200, and 400 years
to quadruple $CO_2$. We would run only one experiment to complete equilibration after ramp up (2500 years) and run the others
only until they lost their sea ice, using the ice-free steady-state run to conduct the three ramp downs. This yields a total of
approximately 6400 simulation years and computational savings by over a factor of 5. Using only three ramping experiments
is sufficient to get an estimate of the equilibrium hysteresis width and location, but the uncertainty of the estimate could still
be high.
Finally, our results indicate that rate-dependent hysteresis and irreversibility of Arctic winter sea ice are expected to be
relevant for realistic rates of $CO_2$ increase. While rate-dependent hysteresis has been explored in other climate contexts (e.g.,
AMOC, Kim et al., 2021; An et al., 2021), previous work on Arctic winter sea ice has typically sought to identify equilibrium
hysteresis in sea ice because it would imply irreversibility of sea ice loss, generally ignoring the out-of-equilibrium behavior
of sea ice under rapid $CO_2$ changes. The SSP585 Scenario in CMIP6 corresponds to a ramping rate of approximately 60 years
per $CO_2$ doubling: a rate at which sea ice in our idealized model already exhibits rate-dependent hysteresis, that is, significant
deviation from its steady state (see Figs. 2 and S2). Since we identify rate-dependent hysteresis in sea ice here in all scenarios,
even without a deep ocean and subsequent recalcitrant warming (Held et al., 2010), we expect rate-dependent hysteresis to be
even more pronounced in GCMs and in the real climate when such long-timescale components are included. We, therefore,
conclude that *on policy-relevant timescales* the significant irreversibility of winter Arctic sea ice involved in rate-dependent
hysteresis is likely to occur in the real climate system due to the expected lagged response regardless of whether an actual
bifurcation (tipping point) in the equilibrium exists.
*Code availability.* An implementation of the Eisenman 2007 sea ice model in python used for this study can be found on Zenodo at:
https://doi.org/10.5281/zenodo.6708812 (Hankel, 2022).
*Author contributions.* CH and ET designed the research project and prepared the manuscript together, CH implemented the model and
conducted the experiments.
*Competing interests.* The authors declare no competing interests.
*Acknowledgements.* The authors would like to thank Ian Eisenman for his helpful input during the project and for the guidance in using
his sea ice model. We thank the anonymous reviewers for their constructive feedback. ET thanks the Weizmann Institute for its hospitality
during parts of this work. This work has been funded by the NSF Climate Dynamics program (joint NSF/NERC) grant AGS-1924538.

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

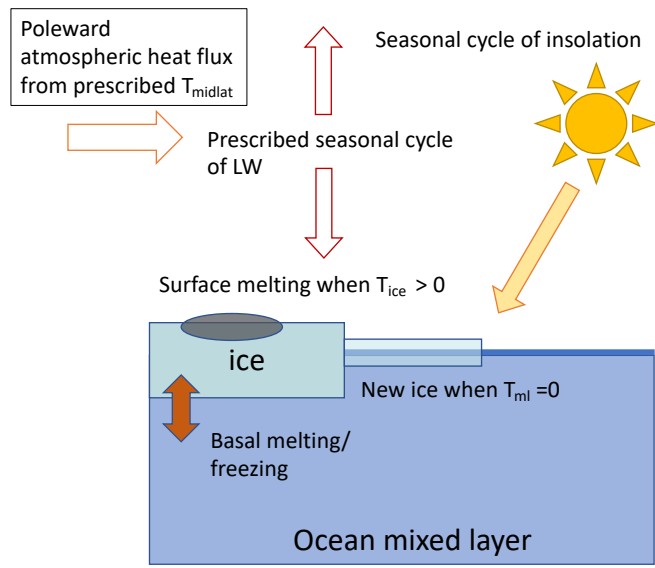

**Figure 1.** Schematic showing some of the key features of the Eisenman (2007) model. Its four prognostic variables are: ice volume, ice area, ice surface temperature, and ocean mixed layer temperature. The full model equations can be found in the Supporting Information.

Park, D.-S. R., Lee, S., and Feldstein, S. B.: Attribution of the recent winter sea ice decline over the Atlantic sector of the Arctic Ocean,
Journal of Climate, 28, 4027–4033, 2015.
Popp, M., Schmidt, H., and Marotzke, J.: Transition to a moist greenhouse with CO2 and solar forcing, Nature communications, 7, 1–10,
436      2016.

Ridley, J., Lowe, J., and Hewitt, H.: How reversible is sea ice loss?, The Cryosphere, 6, 193, 2012.
Ritchie, P. D., Clarke, J. J., Cox, P. M., and Huntingford, C.: Overshooting tipping point thresholds in a changing climate, Nature, 592,
517–523, 2021.
Stroeve, J. and Notz, D.: Changing state of Arctic sea ice across all seasons, Environmental Research Letters, 13, 103 001, 2018.
Stroeve, J., Serreze, M., Drobot, S., Gearheard, S., Holland, M., Maslanik, J., Meier, W., and Scambos, T.: Arctic sea ice extent plummets in
2007, Eos, Transactions American Geophysical Union, 89, 13–14, 2008.
Stroeve, J. C., Serreze, M. C., Holland, M. M., Kay, J. E., Malanik, J., and Barrett, A. P.: The Arctic's rapidly shrinking sea ice cover: a
research synthesis, Climatic change, 110, 1005–1027, 2012.
Strogatz, S.: Nonlinear dynamics and chaos, Westview Press, 1994.
Tredicce, J. R., Lippi, G. L., Mandel, P., Charasse, B., Chevalier, A., and Picqué, B.: Critical slowing down at a bifurcation, American Journal
of Physics, 72, 799–809, 2004.
Wagner, T. J. and Eisenman, I.: How climate model complexity influences sea ice stability, Journal of Climate, 28, 3998–4014, 2015.

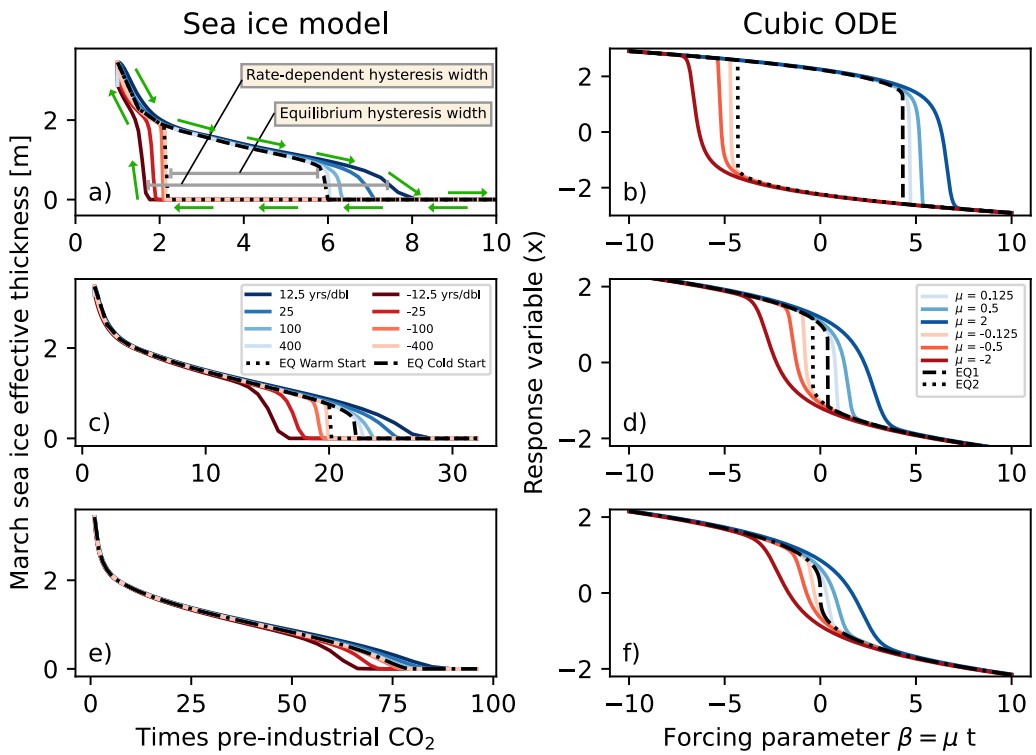

**Figure 2.** Hysteresis runs (time-changing forcing) and equilibrium runs (fixed forcing) for average March sea ice effective thickness (sea ice volume divided by area of the grid cell; panels a,c,d) and the simple ODE from Eq. 1 (b,d,f). The first row corresponds to Scenario 1 (wide bi-stability), the second row to Scenario 2 (narrow bi-stability), and the third to Scenario 3 (no bi-stability). Blue lines indicate simulations with increasing forcing ($CO_2$ or $\beta$), while red lines indicate simulations with decreasing forcing. Dashed and dotted black lines indicate the steady-state values of sea ice or the ODE variable $x$. These two black lines are different when the two initial conditions evolve to two different steady-states. The legends indicate the different ramping rates (represented by darker colors for faster rates), which are in units of years per $CO_2$ doubling in the case of the sea ice model. The green arrows demonstrate the direction of evolving sea ice effective thickness during the hysteresis experiments.

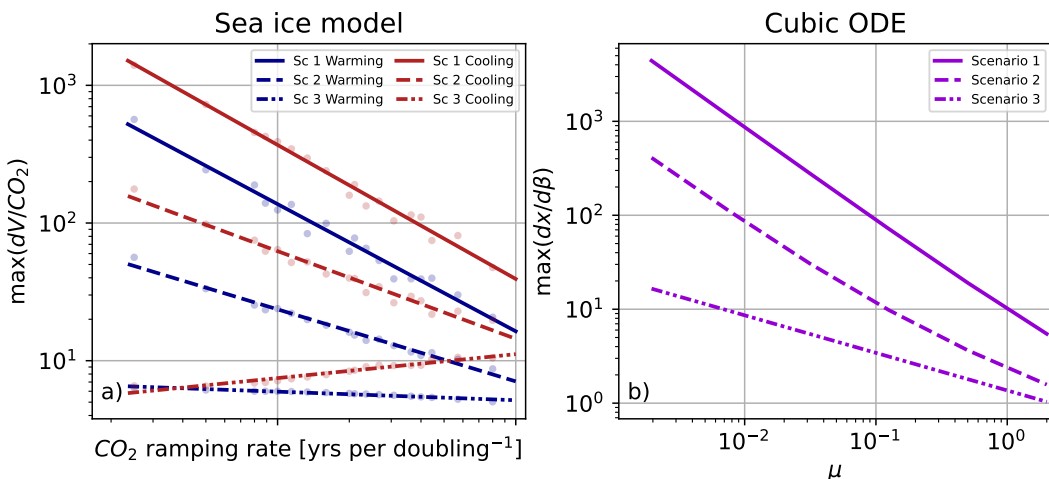

**Figure 3.** Maximum gradient of sea ice effective thickness with respect to $CO_2$ in panel a, and the maximum gradient of $x$ with respect to the forcing parameter $\beta$ in panel b during transient simulations. For the sea ice model (a) the data points from the 18 different runs are shown as faded points, with a superimposed line of best fit. For the cubic ODE (b) the maximum gradient lines corresponding to increasing and decreasing forcing time series are identical due to the symmetry around $\beta = 0$ seen in Fig. 1b, d, and f.

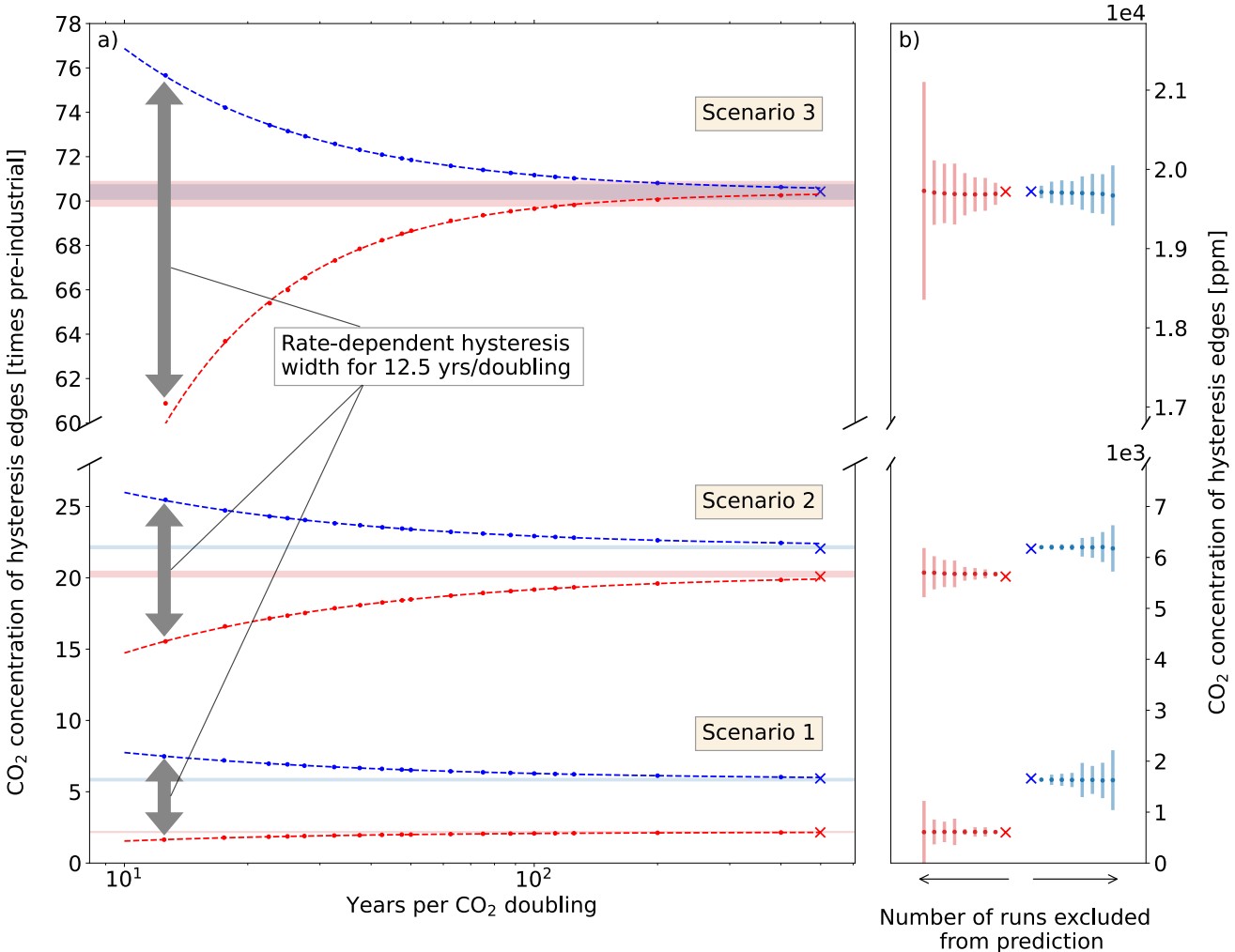

**Figure 4.** Estimating the equilibrium tipping point value from the rate-dependent hysteresis runs. In panel a, the scatter points show the $CO_2$ value of the right and left edges of the rate-dependent hysteresis ($CO_2^i$ and $CO_2^d$, located along increasing (blue) and decreasing (red) $CO_2$ time-series respectively) for different ramping rates. The dashed lines show the curve that is fitted to the scatter points, and the shaded blue and red bands show $\pm 2\sigma$ around the predicted values of $CO_2^i$ and $CO_2^d$ at infinitely slow ramping rates. The blue and red $\times$'s show the true equilibrium values of $CO_2^i$ and $CO_2^d$ (calculated from the fixed $CO_2$ runs starting with cold and warm initial conditions respectively). In panel b, we analyze the accuracy of this prediction as we use fewer transient runs. For the three scenarios, we show the result of sequentially excluding the most gradual ramping simulations from the curve-fitting process used for predictions. The dots and the corresponding bars represent the predicted equilibrium values of $CO_2^i$ and $CO_2^d$, and $\pm 2\sigma$ around the prediction, and dots moving away from the true value with larger error bars correspond to excluding more and more runs from the calculation.