# Peer review of "An approach for projecting the timing of abrupt winter Arctic sea ice loss"

_EGUsphere, 2022_

## Author Response (AR1)

HARVARD UNIVERSITY
DEPARTMENT OF EARTH AND PLANETARY SCIENCES AND
SCHOOL OF ENGINEERING AND APPLIED SCIENCES
20 OXFORD ST, CAMBRIDGE, MASSACHUSETTS, 02138
camille_hankel@g.harvard.edu
Camille Hankel, Eli Tziperman

April 21, 2023

Dr. Stefano Pierini
Editor, *Nonlinear Processes in Geophysics*

**RE: "An approach for projecting the timing of abrupt winter Arctic sea ice loss"**

Dear Dr. Pierini,

Please find below our response to each reviewer's feedback, with references to our updated manuscript. We found both reviewers' comments extremely helpful and have made significant revisions to our manuscript that we feel have clarified the scope and contribution of our project and have strengthened our analysis. In particular, we focused on: 1) better situating our analysis in the dynamical systems literature, clearly establishing where we draw on previous work and where we make new contributions, 2) providing more in-depth discussion on the applicability of our novel method to GCMs and other climate problems, and 3) clarifying the use and definitions of terms such as "hysteresis" and "tipping point" and eliminating the use of "transient hysteresis" (following the suggestion of Reviewer 1), such that the nonlinear processes discussed in this work are more precisely described. We have also further emphasized the main novel contribution of this work: a method that can estimate the existence and location of tipping points without running a model to steady-state.

We thank the reviewers again for taking the time to provide such detailed and constructive feedback.

Sincerely,

Camille Hankel, Eli Tziperman

**Reviewer #1:**

Summary: This paper is concerned with the potential of crossing a bifurcation point in the transition from a seasonally ice-covered to a perennially ice-free Arctic Ocean (loss of winter sea ice), and more broadly with the detection and quantification of hysteresis in a dynamical system. This is a topic that has been debated for a while by the cryospheric science community and features interesting nonlinear processes. Therefore I would assess it in general germane to the journal. The paper is well written, soundly structured, and clearly illustrated. However, from my reading it suffers from a couple of substantial shortcomings at this point that - in my opinion - would have to be remedied before the manuscript is considered for publication.

Thank you for these comments and for the specific helpful suggestions that are now addressed in the new manuscript, as described below. We clarified that our goal is to estimate the equilibrium hysteresis without having to run a climate model to a steady state repeatedly, and we hope this addresses many of the issues raised.

I believe the term "rate-dependent hysteresis" is more widely used and more suitable than "transient hysteresis" and I will use it here. I think a source of confusion in the manuscript is that the concepts of rate-dependent hysteresis and rate-independent hysteresis (ie., the loop traced by equilibrium states) are not separated clearly enough. It is well known, but somehow muddled at points in the manuscript that there is two distinct types of hysteresis at play: one due to bistability in the system and one due to a transient lagged response (which is the result of the inertia present in any physical system).

Thank you for pointing out this confusion in the terminology. We now use "rate-dependent-hysteresis" instead of "transient hysteresis," as suggested:

> Equilibrium hysteresis refers here to the path-dependent solution of a variable due to bi-stability and a bifurcation in the steady-state (in other words, the loop traced by the steady-state solutions). The term "rate-dependent hysteresis" (An et al., 2021; Manoli et al., 2020) describes hysteresis loops that appear in time-changing forcing runs (rather than in the steady state) and that depend on the rate of forcing change. In our analysis "rate-dependent hysteresis" applies to both systems with and without equilibrium hysteresis: it refers to any differences in the results for increasing vs. decreasing $CO_2$ simulations of sea ice that are altered by the rate of $CO_2$ change.

We also now highlight this distinction in our Results:

> In all scenarios, the experiments run with time-changing $CO_2$ exhibit rate-dependent hysteresis; the hysteresis width (lower horizontal gray bar in Fig. 2a) is larger for faster ramping rates (Figs. 2a,c,e). For Scenarios 1 and 2, which have a region of bi-stability (upper gray bar in Fig. 2), this corresponds to a widening from the equilibrium hysteresis (that would

exist even with infinitely slow ramping rates), while in Scenario 3, this hysteresis occurs only in transient simulations and is due to the inertia in the system (the sea ice can't respond instantaneously to forcing changes).

There are well-established methods to probe either. The authors mention some of them but somehow find them lacking. Naturally, the steady-state solutions of a dynamical system can only be approximated in integrations with finite time steps, with the time steps having to become infinitesimal in order to approach the steady state. Yet, whether a solution is approximately converged can be tested in fairly straightforward and well-established ways: for example, as the authors mention, you can take a given fixed forcing level and choose two initial conditions on either side of the suspected stable states (say, very warm and very cold) and run the simulations until you see whether they converge on the same state or not. This avoids complications from the lagged response of the system due to gradually ramping the forcing, which is a separate issue. However, you can typically use the fixed-forcing method to assess how far from the steady-state solution you are in a transient simulation.

We agree that it is well-known how to probe both rate-dependent and equilibrium hysteresis through the fixed-forcing method. We make the case, though, that if the goal is to study the equilibrium hysteresis in a GCM, a fixed-forcing method is too expensive. Thus, our goal is to suggest an alternative method that can be used to deduce the equilibrium hysteresis of Arctic sea ice more efficiently.

We emphasize the motivation for this study in the modified lines 45-52:

> However, Li et al. (2013) further integrated two apparently bi-stable points and found that they equilibrated to the same value of winter sea ice: there was no "true" bi-stability at these two $CO_2$ concentrations, the sea ice was simply out of equilibrium with the $CO_2$ forcing. This calls into question the current use of time-changing $CO_2$ runs to study the bifurcation structure of sea ice.
> In light of the difficulties in using climate model runs with time-changing $CO_2$ (hereafter "transient runs"), the first goal of this work is to understand the relationship between these transient runs and the steady-state value of sea ice in systems with and without bifurcations (since the existence of a bifurcation in winter sea ice remains unknown), and the second goal is to develop a new efficient method for the identification of tipping points from transient runs.

We also now discuss the computational costs of a transient simulation combined with fixed forcing experiments explicitly in the Discussion section:

> We can estimate the efficiency of the proposed approach over more standard ones when applied in a GCM. Taking the experimental setup of Li et al. (2013) as a guide, we can assume that a slow-ramping experiment to $4\times CO_2$ requires a 2000-year ramp up and ramp down with at minimum

a 2500-year equilibration period after each ramp (though they actually allowed the model to equilibrate for nearly 6000 years). Within the 500 ppm width of the rate-dependent hysteresis found by Li et al. (2013), ten fixed-forcing experiments 2500 years long would be needed to test for bi-stability and estimate the tipping point location at a relatively crude accuracy of 100 ppm. This leads to a total of 34,000 simulation years. On the other hand, if we used our proposed approach, we could run three ramping experiments with fast to intermediate rates of 100, 200, and 400 years to quadruple $CO_2$. We would run only one experiment to complete equilibration after ramp up (2500 years) and run the others only until they lost their sea ice, using the ice-free steady-state run to conduct the three ramp downs. This yields a total of approximately 6400 simulation years and computational savings by over a factor of 5. Using only three ramping experiments is sufficient to get an estimate of the equilibrium hysteresis width and location, but the uncertainty of the estimate could still be high.

Of course, you can consider the maximum steepness of the hysteresis loop and how it depends on the rate of forcing change (as the authors propose) to assess whether the hysteresis is rate-dependent or not. You could also use a method (which seems to me equivalent and more common) where you consider the range of forcing over which the hysteresis stretches (length of the gray bars in Fig 1a), again as a function ramping rate. If the ramping rate goes to zero and the hysteresis width disappears, you'll conclude that the hysteresis is purely rate-dependent and the system is not bistable.

We followed the reviewer's suggestion and used the same curve-fitting process on hysteresis width rather than on $CO_2^i$ and $CO_2^d$ separately to extrapolate to infinitely slow ramping rates. The results are satisfactory and are discussed in the Supporting Information and shown in Figure S10:

Another approach for inferring the equilibrium structure of sea ice from transient runs only would be to analyze only the difference between $CO_2^i$ and $CO_2^d$ (i.e., the hysteresis width) as a function of the ramping rate instead of the two values separately and fit a curve to see if this width approaches zero (no bi-stability) at infinitely slow ramping rates. We perform this analysis in Figure S10 and find that it successfully identifies bi-stability in the Scenarios. However, unlike the method in Fig. 4, this method does not provide any prediction of the $CO_2$ value of the tipping point (in Scenarios 1 and 2), so we suggest it is used in addition to, but not in place of, the method in the main text.

We note that letting the ramp rate go to zero, as suggested by the reviewer, would not be computationally feasible in a GCM, and our goal is, therefore, to estimate the equilibrium hysteresis width, if any, more without using infinitely slow ramping

rates or many fixed-forcing runs.

The method proposed here as I understand it from Fig 3a combines two established approaches: (1) you run your simulations very slowly with different initial conditions in order to find the quasi-static approximation to the steady-state hysteresis loop (or in the case of Fig 3, the location of the bifurcations in Scenarios 1 and 2). (2) You run it faster and look at how the width of the hysteresis loop depends on the rate of forcing change in order to assess the rate-dependence of the hysteresis loop.

Yes, one of our goals is to assess if the rate-dependent hysteresis reflects an equilibrium hysteresis. But our main goal is to estimate the width and location of an equilibrium hysteresis, thereby predicting the location of sea ice tipping points, without having to use a quasi-static approximation with requires a very long, infeasible GCM run. We now explain this more clearly in the abstract and introduction.

While some of the ramping rates we use for demonstrating the method are relatively gradual, we emphasize that the approach itself does not rely on using a very slow ramping rate to approach a quasi-steady state (the first approach listed by the reviewer above), but rather just uses the convergence behavior of several runs with different ramping rates to estimate the width of the bi-stability and the $CO_2$ value of the most abrupt ice change. We now make sure that the first point is clear in our discussion of Fig. 4b,

> The predicted values of $CO_2^i$ and $CO_2^d$ are remarkably accurate for all scenarios (points approaching the red and blue $\times$ in Fig. 4b), even when excluding several of the slower ramping experiments. This is an important test because when this method is applied to a GCM, one would only have a smaller number of faster ramping experiments due to computational limitations.

As far as I understand it, this combined approach may be OK, but I see two issues: (A) I feel it confounds the two issues of rate-independent and rate-dependent hysteresis, rather than separate them clearly. (B) The authors fail to show that this approach is actually computationally advantageous, compared to the standard methods that probe either 1) or 2). How would this proposed approach be implemented in comprehensive GCMs?

We hope that our revised careful definitions of "equilibrium hysteresis", and "rate-dependent hysteresis", now in lines 177–182 help separate these issues. In addition, we add clarifying language to highlight that one of our goals is precisely to identify the equilibrium hysteresis from efficient rate-dependent experiments:

> Our main novel result, presented next, is a method for finding the $CO_2$ concentration at which a bifurcation (if any) occurs in the equilibrium using computationally feasible transient model runs instead of fixed-forcing steady-state runs. We are interested in this $CO_2$ concentration because it determines the threshold beyond which significant sea ice loss is practically irreversible (Ritchie et al., 2021). In our simple, inexpensive model, we can test the estimates of the bi-stability and associated tipping points derived from transient model runs against the known true tipping points and equilibrium structure that are found from fixed-forcing runs (see Methods). When used in a GCM, our method would provide a prediction for the existence and location of tipping points when the equilibrium value of sea ice is actually unknown. Thus, this section is a proof of concept that our new method can accurately determine whether observed rate-dependent hysteresis is caused by lag around a system with no bi-stability or tipping points or caused by a rate-dependent widening of an equilibrium hysteresis loop in a system with tipping points.

Following the reviewer's suggestion, we now add in the Discussion (lines 323–334) a calculation of the estimated cost for a combined slow ramping plus fixed forcing method (i.e., if Li et al. (2013) did additional fixed-forcing experiments to conclusively find/rule out bi-stability) vs. our multiple ramping experiments method to show that our method would be computationally advantageous, as mentioned above in response to the reviewer's earlier comment.

It is further unclear to me what is gained by fitting a seemingly ad-hoc exponential decay equation to match the data points in Fig 3a.

We hope the revised manuscript and the above answers (in particular lines 164–170 and 255–261) now make it clearer that our goal is to estimate the equilibrium structure based on rate-dependent experiments. This requires a fit to the simulations in order to extrapolate to infinitely slow ramping rates that would represent the equilibrium behavior but can not be directly simulated in an efficient way.

Some of the lack of clarity here is compounded by ambiguity in the use of the term tipping point: at points it seems that tipping point refers to a bifurcation, at other points it is suggested that the system can feature a tipping point without bifurcation (e.g., l.25).

Thank you for pointing out the lack of clarity which we now corrected. We now say in lines 16–25 in the Introduction that abrupt loss of sea ice may or may not be governed by irreversible processes and an associated bifurcation, and that such loss is referred to as a "tipping point" only when this is the case. We are careful to use this terminology correctly in the discussion of Fig. 4 as well

Abrupt loss of Arctic sea ice could be driven by local positive feedback mechanisms (Curry et al., 1995; Abbot and Tziperman, 2008; Abbot et al., 2009; Kay et al., 2012; Leibowicz et al., 2012; Burt et al., 2016; Feldl et al., 2020; Hankel and Tziperman, 2021), remote feedback mechanisms that increase heat flux from the mid-latitudes (Holland et al., 2006; Park et al., 2015), or by the natural threshold corresponding to the seawater freezing point (Bathiany et al., 2016). If such an abrupt loss is caused

by irreversible processes (typically, strong positive feedback mechanisms as opposed to the reversible mechanism of a freezing point threshold of Bathiany et al., 2016) it is referred to here as a "tipping point". A tipping point in the sense used here is understood mathematically as a change in the number or stability of steady-state solutions (Ghil and Childress, 1987; Strogatz, 1994) as a function of $CO_2$ and is also known as a "bifurcation". We note that some of the climate literature uses "tipping points" in a more general sense of a relatively rapid change (e.g., Lenton, 2012).

Later also, we have revised our wording to say:

> Furthermore, three out of seven fully-complex Global Climate Models (GCMs) that lost their winter sea ice completely in the CMIP5 Extended RCP8.5 Scenario showed a very abrupt change in winter Arctic sea ice resembling a tipping point (Hezel et al., 2014; Hankel and Tziperman, 2021).

Furthermore, the last paragraph seems to misrepresent the state of scientific debate somewhat: On the one hand, yes, most previous work on this topic is focused on whether or not there is rate-independent hysteresis. On the other hand, I'm fairly certain most authors would readily agree that there would be rate-dependent hysteresis if the climate was warmed quickly and subsequently cooled, simply due to the inertia of the Arctic and the global climate system. Most people wouldn't call this hysteresis I would guess, but rather a lagged response.

Thanks for this comment. We revised this last paragraph (and in particular the last sentence discussed by the reviewer) to refer explicitly to rate-dependent hysteresis due to a lagged response rather than hysteresis,

> We, therefore, conclude that *on policy-relevant timescales* the significant irreversibility of sea ice involved in rate-dependent hysteresis is likely to occur in the real climate system due to the expected lagged response regardless of whether an actual bifurcation (tipping point) in the equilibrium exists.

Finally, the authors state that they find "transient hysteresis" in all scenarios, and again this mixes the two distinct hysteresis concepts. They really find rate-independent hysteresis in scenarios 1 and 2, and no bistability but a lagged response in scenario 3.

We essentially agree, and we now use "rate-dependent" instead of transient, and clarify our findings in each scenario using this corrected terminology. Please see lines 188–191 in the revised manuscript, as already quoted in a response to a previous comment.

My second main concern is regarding the applicability to the case of winter sea ice. For one, it is unexpected that the authors use the Eisenman (2007) which is from a non-peer reviewed WHOI summer school report. It is my understanding that subsequent versions developed by Eisenman and Wettlaufer (PNAS, 2009) and Eisenman (JGR, 2012) are better formulated variants of this model and would seem a more natural starting point. It seems at least appropriate to justify the use of this model over the later peer-reviewed versions. (I would also urge the authors to summarize the main equations of the model in the main text, in order to make the article more self-contained.)

We now justify the use of the earlier Eisenman model in the revised text, and emphasize that we expect all of our main results to hold even if we switched to a later (further simplified) version of the model:

> Subsequent versions of this sea ice model have been used in Eisenman and Wettlaufer (2009), Eisenman (2012), and Wagner and Eisenman (2015). Those versions are derived from the model used here, making a few further modest simplifications (using a hyperbolic tangent function for surface albedo, assuming the ice surface temperature is in a steady state, combining all prognostic variables into one, enthalpy) that do not affect the qualitative behavior of the model (i.e., the nature of summer and winter sea ice bifurcations). We choose to implement the earlier version of the model because it explicitly represents the key physical variables of ice volume, area, ocean temperature, and ice temperature as prognostic variables — as opposed to combining them all into a single enthalpy — and thus provides more transparency and interpretability. We, therefore, do not expect our results to change if we use any of the later model versions.

Thank you for suggesting we take steps to make the paper more self-contained. To achieve this, we now add a few sentences to our model description explaining the key thermodynamic mechanisms of the model and add a schematic Fig. 1 that gives a general overview of the model features:

> Sea ice growth and loss are primarily determined by the heat budget at the bottom of the ice and are therefore set by the balance between ocean-ice heat exchanges, and heat loss through the ice to the atmosphere. When conditions for surface melting are met (when the ice surface temperature is zero and net fluxes on the ice are positive), all surface heating goes into melting ice and the surface albedo of the ice is set to the melt pond albedo. The ocean temperature is affected by shortwave and longwave fluxes in the fraction of the box that is ice-free, and by ice-ocean heat exchanges. When the ocean temperature reaches zero, all additional cooling goes into ice production while the ocean temperature remains constant.

Maybe more importantly, the main story of this paper does not seem to be concerned with Arctic sea ice, but rather with testing hysteretic behavior in dynamical systems. In this regard, the placing of the findings in context of existing literature is lacking: for example, the system of eq 1 has been used to study related questions in many cases, none of which are cited here, which makes it almost sound like the authors suggest this is an original contribution. To mind come the classic textbook by Strogatz ("Nonlinear Dynamics and Chaos", see Ch. 3.6.), Ditlevsen and Johnsen "Tipping points: Early warning and wishful thinking." Geophysical Research Letters 37.19 (2010), and recently Boers "Observation-based early-warning signals for a collapse of the Atlantic Meridional Overturning Circulation." Nature Climate Change 11.8 (2021): 680-688. In the latter text you'll see in Fig 1 that the transient solution continues beyond the bifurcation point before the abrupt transition, just as discussed here.

Thank you for providing these suggestions for better contextualizing our findings. While continuation beyond a tipping point due to rate-dependent hysteresis has not been discussed in the context of Arctic sea ice previously, we agree that it could have been anticipated from the previous literature you mention dealing with other climate systems (AMOC, etc.). Similarly, we now cite other works that used this standard cubic dynamical system to put our study in the context of related climate literature. See lines 202–222 of the Results section:

> Previous work in the dynamical systems literature (e.g., Haberman, 1979; Mandel and Erneux, 1987; Baer et al., 1989; Breban et al., 2003; Tredicce et al., 2004; Kaszás et al., 2019) has examined a variety of simple systems to understand the nature of bifurcations in the presence of a time-changing ("drifting" or "transient") forcing parameter. In the climate literature as well (e.g., Ditlevsen and Johnsen, 2010; Bathiany et al., 2018; Ritchie et al., 2021; Boers, 2021), idealized dynamical systems similar to our Eqn. 1 have been used to understand the predictability of tipping points in the presence of noise, and the ability to recover from such tipping points ("overshoot" scenarios). These works, as well as the AMOC study of An et al. (2021), found that a system with a bifurcation that is run with a time-changing forcing parameter can follow a given equilibrium value beyond the bifurcation value of the forcing parameter before undergoing the tipping point transition to the new equilibrium value. This is consistent with the out-of-equilibrium behaviors we find for sea ice in Scenarios 1 and 2. To our knowledge, the simple ODE used here has not yet been analyzed with our specific goal in mind: to compare the shape of rate-dependent hysteresis loops in generic dynamical systems both with and without bifurcations, and to address the question of whether the equilibrium behavior can be inferred from the rate-dependent behavior of such systems.
>
> To address these two goals, we configure Eqn. 1 analogously to the sea ice model in three scenarios with wide bi-stability (Scenario 1), narrow

bi-stability (Scenario 2), and no bi-stability (Scenario 3) and force it with a time-changing forcing parameter. In Figs. 2b,d,f, we see that the three scenarios with similar dynamics (but different equilibrium structures) all display rate-dependent hysteresis, similar to the result from the sea ice model. Specifically, even when there is only one stable equilibrium solution in both models (Scenario 3, panels e and f), there is still a narrow region of rate-dependent hysteresis. Thus, we find that the inability to tell if rate-dependent hysteresis in Arctic winter sea ice is accompanied by an underlying equilibrium hysteresis appears to be a generic feature of dynamical systems, which helps explain the challenges of interpreting the results of Li et al. (2013).

In our Discussion section, we are careful to identify our original contributions with respect to the dynamical systems literature:

We demonstrated that the transient sea-ice responses under a time-changing $CO_2$ reflect the generic behavior of a nonlinear dynamical system (e.g., our Eqn. 1): specifically, we showed that systems with and without bi-stability can also produce qualitatively indistinguishable rate-dependent hysteresis behavior.

In addition, as discussed above and as is hopefully now clear in the revised manuscript, we emphasize that the main contribution of this work is our new method that can predict the equilibrium behavior of a system from time-changing forcing runs alone, making identification of tipping points in global climate models computationally feasible.

We agree with the reviewer's comment that while we developed this method in the context of sea ice, the method itself is quite general to non-autonomous dynamical systems. We now highlight one advantage of this generality in our Discussion, lines 317–318:

The generality of the method also highlights another advantage: the same set of ramping experiments in a GCM could be used to analyze all suspected tipping elements in the Earth's climate system simultaneously.

**Reviewer #2:** Summary: In this paper, Camille Hankel and Eli Tziperman present some interesting results regarding the estimate of tipping-point behaviour in a simplified 1-D sea-ice model, with potential applicability for the analysis of GCM results. Overall, I enjoyed reading this paper, and think it certainly has great potential for publication in this journal. In particular, obtaining a method that allows one to more readily estimate and understand the potential hysteresis-behaviour of more complex models would be highly welcome. I think this aim could eventually be reached, but in my view the following overarching points need to be addressed first:

Thank you for this summary and for the specific suggestions, which, as described below, are addressed in the revised manuscript.

1. While I enjoyed reading the conceptual framework of estimating potential tipping point behaviour of sea ice, it did not become fully clear to me which part of this analysis is novel also in the broader context of dynamical-system analysis, and which part is primarily an application of known concepts to the case of sea ice. Either cases of novelty would certainly be fine, and welcome, but it'd be helpful to have more information on which part of the paper falls into which category.

Thank you for pointing this out. We now clarify in several places that the novel element of our work is the development of an efficient method that allows us to estimate the existence and location of tipping points from inexpensive time-changing forcing runs alone. The cubic ODE is used to demonstrate the generality of the issue that makes such a method necessary, and that the proposed method works for such a generic ODE as well and therefore is not limited to the sea ice model used. We explain,

> Previous work in the dynamical systems literature (e.g., Haberman, 1979; Mandel and Erneux, 1987; Baer et al., 1989; Breban et al., 2003; Tredicce et al., 2004; Kaszás et al., 2019) has examined a variety of simple systems to understand the nature of bifurcations in the presence of a time-changing ("drifting" or "transient") forcing parameter. In the climate literature as well (e.g., Ditlevsen and Johnsen, 2010; Bathiany et al., 2018; Ritchie et al., 2021; Boers, 2021), idealized dynamical systems similar to our Eqn. 1 have been used to understand the predictability of tipping points in the presence of noise, and the ability to recover from such tipping points ("overshoot" scenarios). These works, as well as the AMOC study of An et al. (2021), found that a system with a bifurcation that is run with a time-changing forcing parameter can follow a given equilibrium value beyond the bifurcation value of the forcing parameter before undergoing the tipping point transition to the new equilibrium value. This is consistent with the out-of-equilibrium behaviors we find for sea ice in Scenarios 1 and 2. To our knowledge, the simple ODE used here has not yet been analyzed with our specific goal in mind: to compare the shape of rate-dependent hysteresis loops in generic dynamical systems both with and without bifurcations, and to address the question of whether the

equilibrium behavior can be inferred from the rate-dependent behavior of such systems.

To address these two goals, we configure Eqn. 1 analogously to the sea ice model in three scenarios with wide bi-stability (Scenario 1), narrow bi-stability (Scenario 2), and no bi-stability (Scenario 3) and force it with a time-changing forcing parameter. In Figs. 2b,d,f, we see that the three scenarios with similar dynamics (but different equilibrium structures) all display rate-dependent hysteresis, similar to the result from the sea ice model. Specifically, even when there is only one stable equilibrium solution in both models (Scenario 3, panels e and f), there is still a narrow region of rate-dependent hysteresis. Thus, we find that the inability to tell if rate-dependent hysteresis in Arctic winter sea ice is accompanied by an underlying equilibrium hysteresis appears to be a generic feature of dynamical systems, which helps explain the challenges of interpreting the results of Li et al. (2013).

We also revise our Discussion to highlight the novel use of the ODE,

> We demonstrated that the transient sea-ice responses under a time-changing $CO_2$ reflect the generic behavior of a nonlinear dynamical system (e.g., our Eqn. 1): specifically, we showed that systems with and without bi-stability can also produce qualitatively indistinguishable rate-dependent hysteresis behavior..

as well as the main contribution, the method to detect tipping points:

> We showed that even in runs with a very slow-changing $CO_2$, the system can be surprisingly far from the equilibrium as it undergoes a tipping point, consistent with the work of Li et al. (2013). In addition, even with a very slow ramping experiment, one would always have to perform additional expensive fixed-forcing experiments (as done by Li et al., 2013) to confirm that the experiment was indeed in quasi-equilibrium. Instead, we propose a novel method that uses a few fast-ramping experiments to efficiently predict the true range of bi-stability and provide uncertainty estimates on this prediction.

2. I am happy to accept the validity of the general framework as outlined here, but I am much less certain that the results obtained here are applicable to the real world. While the authors are careful in not over-emphasising their results in this regard, I think that a broader discussion of the applicability of the Eisenman-model for the analysis of real sea ice seems warranted, in particular in light of the (in my view excellent) study by Wagner and Eisenman, 2015 (https://doi.org/10.1175/JCLI-D-14-00654.1). How do the limitations of the employed model affect the general validity of the results obtained here?

Thank you for suggesting this, we now address these issues in the manuscript in a

few ways, listed below.

First, we now explicitly mention the Wagner and Eisenman (2015) work in the Introduction, highlighting how it contributes to, but does not resolve, the debate over the existence of a winter sea ice tipping point:

> Wagner and Eisenman (2015) showed that a winter tipping point disappeared from a simple model of sea ice with no active atmosphere when a longitudinal dimension was added. On the other hand, other literature (e.g., Abbot and Tziperman, 2008; Hankel and Tziperman, 2021) has demonstrated the importance of atmospheric feedbacks, not included in the model of Wagner and Eisenman (2015), in inducing winter sea ice tipping point. Furthermore, three out of seven fully-complex Global Climate Models (GCMs) that lost their winter sea ice completely in the CMIP5 Extended RCP8.5 Scenario showed a very abrupt change in winter Arctic sea ice resembling a tipping point (Hezel et al., 2014; Hankel and Tziperman, 2021).

Second, we highlight how our work fits into the goal of understanding sea ice tipping points in the *real* climate, writing in the Introduction:

> Finally, we propose a novel approach for uncovering the underlying equilibrium behavior — and thus the existence of tipping points — in comprehensive models where it is computationally infeasible to simulate steady-state conditions for many $CO_2$ values. We emphasize the importance of such a method given the model-dependent nature of winter sea ice tipping points discussed above; uncovering the existence of sea ice tipping points in GCMs, which are the most realistic representation of Arctic-wide sea ice behavior that we have, is the next step toward understanding whether such tipping points exist in the real climate system.

Finally, we now discuss the general applicability of this tipping point identification framework, also in response to the reviewer's comment below, noting that we expect our method to work across different dynamical systems experiencing time-changing forcing:

> We demonstrated that the method we propose can accurately predict the steady-state behavior of sea ice in a simple model; now we discuss applying this method to a GCM. First, we note that while we use a highly idealized model of sea ice in this study, the method developed deals with identifying bi-stability in complex systems with unknown equilibrium structures more generally. This means that the framework should be applicable to other models (including GCMs), since moving from fast to slower ramping rates allows convergence to the equilibrium behavior. It could also be used in the context of different climate problems, for example, in identifying the abrupt transitions to a moist greenhouse (Popp

et al., 2016), runaway greenhouse (Goldblatt et al., 2013), or snowball Earth state (Hyde et al., 2000). The functional form used to fit the transient runs, as well as the level of certainty achieved from a given number of experiments, would likely depend on the given model and climate problem analyzed. Possible challenges in finding the functional best fit to the transient runs might mirror those of Gregory et al. (2004) who encountered difficulties when trying to fit a line to un-equilibrated GCM runs with a different goal of deducing the equilibrium climate sensitivity. We suggest that a careful examination of the residuals from a given fit can help guide the choice of functional form.

The generality of the method also highlights another advantage: the same set of ramping experiments in a GCM could be used to analyze all suspected tipping elements in the Earth's climate system simultaneously. The main challenge we anticipate in applying this method to GCMs comes from the significant stochastic variability and multiple timescales of forcings that may render the calculated width of the rate-dependent hysteresis more uncertain in a GCM. Nonetheless, using multiple runs to estimate the width of the bi-stability of a given climate variable and providing a quantified uncertainty on such a prediction should offer a potential improvement over using a single hysteresis experiment.

3. I think it'd be very helpful for a geo-physical audience if the distinction between "abrupt changes" and "hysteresis" would be made more explicit. Not all tipping points need to be "abrupt" (depending on the definition of this term) and not all abrupt changes indicate hysteresis behaviour. For example, I would assume that many of the "abrupt changes" in winter sea ice in GCMs as analysed in Bathiany et al., 2016 are fully reversible if the argumentation of their paper is correct. Being clear upfront as to which of these different dynamic behaviours are studied here would help, in particular given that these concepts are sometimes a bit muddled in the literature.

Thank you for pointing this out; we have clarified our definition of "abrupt changes" vs "tipping point" in the Introduction, and have defined "equilibrium hysteresis" and "rate-dependent hysteresis" as the terms we now use in the Results.

In the Introduction, we now write:

Abrupt loss of Arctic sea ice could be driven by local positive feedback mechanisms (Curry et al., 1995; Abbot and Tziperman, 2008; Abbot et al., 2009; Kay et al., 2012; Leibowicz et al., 2012; Burt et al., 2016; Feldl et al., 2020; Hankel and Tziperman, 2021), remote feedback mechanisms that increase heat flux from the mid-latitudes (Holland et al., 2006; Park et al., 2015), or by the natural threshold corresponding to the seawater freezing point (Bathiany et al., 2016). If such an abrupt loss is caused by irreversible processes (typically, strong positive feedback mechanisms as opposed to the reversible mechanism of a freezing point threshold of

Bathiany et al., 2016) it is referred to here as a "tipping point". A tipping point in the sense used here is understood mathematically as a change in the number or stability of steady-state solutions (Ghil and Childress, 1987; Strogatz, 1994) as a function of $CO_2$ and is also known as a "bifurcation". We note that some of the climate literature uses "tipping points" in a more general sense of a relatively rapid change (e.g., Lenton, 2012).

and we have modified the following sentence to be consistent with our definition of a tipping point:

Furthermore, three out of seven fully-complex Global Climate Models (GCMs) that lost their winter sea ice completely in the CMIP5 Extended RCP8.5 Scenario showed a very abrupt change in winter Arctic sea ice resembling a tipping point (Hezel et al., 2014; Hankel and Tziperman, 2021).

To reiterate the definition of "hysteresis" used here we now write in the Results:

Equilibrium hysteresis refers here to the path-dependent solution of a variable due to bi-stability and a bifurcation in the steady-state (in other words, the loop traced by the steady-state solutions). The term "rate-dependent hysteresis" (An et al., 2021; Manoli et al., 2020) describes hysteresis loops that appear in time-changing forcing runs (rather than in the steady state) and that depend on the rate of forcing change. In our analysis "rate-dependent hysteresis" applies to both systems with and without equilibrium hysteresis: it refers to any differences in the results for increasing vs. decreasing $CO_2$ simulations of sea ice that are altered by the rate of $CO_2$ change.

4. Some reference to the common concepts of obtaining effective climate sensitivity in 4x CO2 simulations from curve fitting etc., seems warranted, given that these concepts are at least to some degree similar to the concepts suggested here. (e.g., https://doi.org/10.1029/2003GL018747)

We are grateful for this idea to mention the Gregory paper, which we agree is very relevant in its aims to our study. We now mention in the introduction,

Our goal has some parallels to that of Gregory et al. (2004), who used un-equilibrated GCM runs to deduce the equilibrium climate sensitivity when fully-equilibrated runs were computationally infeasible.

and in the conclusions,

Possible challenges in finding the functional best fit to the transient runs might mirror those of Gregory et al. (2004) who encountered difficulties

when trying to fit a line to un-equilibrated GCM runs with a different goal of deducing the equilibrium climate sensitivity.

5. Given that this study is framed as a proof of concept for the applicability in GCMs (at least this is what I perceive as the overall framing, for example in the abstract), I think more discussion is needed on the applicability of this idea. For example, which equilibrium states would you suggest for the CO2 increase/ decrease runs? How do other ESM-components affect the applicability of this method? Should parts of the ESM be held fixed? Would this method allow one to study equilibrium of all major ESM components that might exhibit tipping point behaviour from the same set of simulations? Is there anything specific about the analysis of sea ice from these simulations?

Regarding the application of this method to a GCM, we have now given some guidelines (based on the results of Li et al., 2013) in our new discussion of the computational efficiency of this method:

> We can estimate the efficiency of the proposed approach over more standard ones when applied in a GCM. Taking the experimental setup of Li et al. (2013) as a guide, we can assume that a slow-ramping experiment to $4{\times}CO_2$ requires a 2000-year ramp up and ramp down with at minimum a 2500-year equilibration period after each ramp (though they actually allowed the model to equilibrate for nearly 6000 years). Within the 500 ppm width of the rate-dependent hysteresis found by Li et al. (2013), ten fixed-forcing experiments 2500 years long would be needed to test for bi-stability and estimate the tipping point location at a relatively crude accuracy of 100 ppm. This leads to a total of 34,000 simulation years. On the other hand, if we used our proposed approach, we could run three ramping experiments with fast to intermediate rates of 100, 200, and 400 years to quadruple $CO_2$. We would run only one experiment to complete equilibration after ramp up (2500 years) and run the others only until they lost their sea ice, using the ice-free steady-state run to conduct the three ramp downs. This yields a total of approximately 6400 simulation years and computational savings by over a factor of 5. Using only three ramping experiments is sufficient to get an estimate of the equilibrium hysteresis width and location, but the uncertainty of the estimate could still be high.

With regards to the question of the specificity of this method and the ability to use the same set of simulations to analyze multiple Earth system tipping elements, we have now addressed this in our revised Discussion section:

> We demonstrated that the method we propose can accurately predict the steady-state behavior of sea ice in a simple model; now we discuss applying this method to a GCM. First, we note that while we use a highly

idealized model of sea ice in this study, the method developed deals
with identifying bi-stability in complex systems with unknown equilib-
rium structures more generally. This means that the framework should
be applicable to other models (including GCMs), since moving from fast
to slower ramping rates allows convergence to the equilibrium behavior.
... The functional form used to fit the transient runs, as well as the
level of certainty achieved from a given number of experiments, would
likely depend on the given model and climate problem analyzed. Possi-
ble challenges in finding the functional best fit to the transient runs might
mirror those of Gregory et al. (2004) who encountered difficulties when
trying to fit a line to un-equilibrated GCM runs with a different goal of
deducing the equilibrium climate sensitivity. We suggest that a careful
examination of the residuals from a given fit can help guide the choice of
functional form.

The generality of the method also highlights another advantage: the
same set of ramping experiments in a GCM could be used to analyze all
suspected tipping elements in the Earth's climate system simultaneously.

Some minor additional comments:

**l. 2:** "Collapse of a sea-ice equilibrium" sounds dramatic, but it is unclear to me
what you mean by this terminology
We changed this to say "disappearance of a sea-ice equilibrium", to be more
consistent with the definition of a bifurcation.

**l.11:** Also winter sea ice has retreated rapidly in recent decades
Thank you for pointing this out, we've now modified the sentence and included
some additional references to observed winter sea ice decline.:

> Sea ice is already exhibiting rapid retreat with warming, especially in
> the summertime, (Comiso and Parkinson, 2004; Nghiem et al., 2007;
> Stroeve et al., 2008; Notz and Stroeve, 2016; Stroeve and Notz, 2018),
> shortening the time that socioeconomic and ecological systems have
> to adapt.

**l.15:** "Abrupt loss or tipping point": These are different concepts, so shouldn't be
merged I think
Thank you for pointing this out, we've now fixed this and more carefully defined
the two terms, as quoted in one of the responses above.

**l.54:** I wonder if this is a major, robust result of this study that should be highlighted
more?
We agree that this is a significant result, consistent with Li et al. (2013) and
more thoroughly examined here, as we now highlight in the Discussion section:

> A consequence of this is that even in very slow runs, the system can
> be surprisingly far from the equilibrium as it undergoes a tipping
> point, consistent with the work of Li et al. (2013).

**l.138:** Should this read Figs. 1a,c,e?

Yes, thank you, fixed.

**l.147:** Duplicate "Li et al"

Fixed, thank you.

**References**

Abbot, D. S. and Tziperman, E. (2008). Sea ice, high-latitude convection, and equable climates. *Geophysical Research Letters*, 35(3).

Abbot, D. S., Walker, C., and Tziperman, E. (2009). Can a convective cloud feedback help to eliminate winter sea ice at high $CO_2$ concentrations? *J. Climate*, 22(21):5719–5731.

An, S.-I., Kim, H.-J., and Kim, S.-K. (2021). Rate-dependent hysteresis of the atlantic meridional overturning circulation system and its asymmetric loop. *Geophysical Research Letters*, 48(1):e2020GL090132.

Baer, S. M., Erneux, T., and Rinzel, J. (1989). The slow passage through a hopf bifurcation: delay, memory effects, and resonance. *SIAM Journal on Applied mathematics*, 49(1):55–71.

Bathiany, S., Notz, D., Mauritsen, T., Raedel, G., and Brovkin, V. (2016). On the potential for abrupt arctic winter sea ice loss. *Journal of Climate*, 29(7):2703–2719.

Bathiany, S., Scheffer, M., Van Nes, E., Williamson, M., and Lenton, T. (2018). Abrupt climate change in an oscillating world. *Scientific reports*, 8(1):1–12.

Boers, N. (2021). Observation-based early-warning signals for a collapse of the atlantic meridional overturning circulation. *Nature Climate Change*, 11(8):680–688.

Breban, R., Nusse, H. E., and Ott, E. (2003). Lack of predictability in dynamical systems with drift: scaling of indeterminate saddle-node bifurcations. *Physics Letters A*, 319(1-2):79–84.

Burt, M. A., Randall, D. A., and Branson, M. D. (2016). Dark warming. *Journal of Climate*, 29(2):705–719.

Comiso, J. C. and Parkinson, C. L. (2004). Satellite observed changes in the arctic. *Physics Today*.

Curry, J. A., Schramm, J. L., and Ebert, E. E. (1995). Sea ice–albedo climate feedback mechanism. *J. Climate*, 8:240–247.

Ditlevsen, P. D. and Johnsen, S. J. (2010). Tipping points: Early warning and wishful thinking. *Geophys. Res. Lett.*, 37(L19703).

Eisenman, I. (2012). Factors controlling the bifurcation structure of sea ice retreat. *Journal of Geophysical Research: Atmospheres*, 117(D1).

Eisenman, I. and Wettlaufer, J. S. (2009). Nonlinear threshold behavior during the loss of arctic sea ice. *Proc Nat Acad Sci USA*, 106:28–32.

Feldl, N., Po-Chedley, S., Singh, H. K., Hay, S., and Kushner, P. J. (2020). Sea ice and atmospheric circulation shape the high-latitude lapse rate feedback. *npj Climate and Atmospheric Science*, 3(1):1–9.

Ghil, M. and Childress, S. (1987). *Topics in Geophysical Fluid Dynamics: Atmospheric Dynamics, Dynamo Theory and Climate Dynamics*. Springer-Verlag, New York.

Goldblatt, C., Robinson, T. D., Zahnle, K. J., and Crisp, D. (2013). Low simulated radiation limit for runaway greenhouse climates. *Nature Geoscience*, 6(8):661–667.

Gregory, J., Ingram, W., Palmer, M., Jones, G., Stott, P., Thorpe, R., Lowe, J., Johns, T., and Williams, K. (2004). A new method for diagnosing radiative forcing and climate sensitivity. *Geophysical research letters*, 31(3).

Haberman, R. (1979). Slowly varying jump and transition phenomena associated with algebraic bifurcation problems. *SIAM Journal on Applied Mathematics*, 37(1):69–106.

Hankel, C. and Tziperman, E. (2021). The role of atmospheric feedbacks in abrupt winter arctic sea ice loss in future warming scenarios. *Journal of Climate*, 34(11):4435–4447.

Hezel, P., Fichefet, T., and Massonnet, F. (2014). Modeled arctic sea ice evolution through 2300 in cmip5 extended rcps. *The Cryosphere*, 8(4):1195–1204.

Holland, M. M., Bitz, C. M., and Tremblay, B. (2006). Future abrupt reductions in the summer arctic sea ice. *Geophysical Research Letters*, 33(23).

Hyde, W. T., Crowley, T. J., Baum, S. K., and Peltier, W. R. (2000). Neoproterozoic 'snowball earth' simulations with a coupled climate/ice-sheet model. *Nature*, 405:425–429.

Kaszás, B., Feudel, U., and Tél, T. (2019). Tipping phenomena in typical dynamical systems subjected to parameter drift. *Scientific reports*, 9(1):8654.

Kay, J. E., Holland, M. M., Bitz, C. M., Blanchard-Wrigglesworth, E., Gettelman, A., Conley, A., and Bailey, D. (2012). The influence of local feedbacks and northward heat transport on the equilibrium arctic climate response to increased greenhouse gas forcing. *Journal of Climate*, 25(16):5433–5450.

Leibowicz, B. D., Abbot, D. S., Emanuel, K. A., and Tziperman, E. (2012). Correlation between present-day model simulation of Arctic cloud radiative forcing and sea ice consistent with positive winter convective cloud feedback. *J. Adv. Model. Earth Syst.*, 4.

Lenton, T. M. (2012). Arctic climate tipping points. *Ambio*, 41(1):10–22.

Li, C., Notz, D., Tietsche, S., and Marotzke, J. (2013). The transient versus the equilibrium response of sea ice to global warming. *Journal of Climate*, 26(15):5624–5636.

Mandel, P. and Erneux, T. (1987). The slow passage through a steady bifurcation: delay and memory effects. *Journal of statistical physics*, 48(5):1059–1070.

Manoli, G., Fatichi, S., Bou-Zeid, E., and Katul, G. G. (2020). Seasonal hysteresis of surface urban heat islands. *Proceedings of the National Academy of Sciences*, 117(13):7082–7089.

Nghiem, S., Rigor, I., Perovich, D., Clemente-Colón, P., Weatherly, J., and Neumann, G. (2007). Rapid reduction of arctic perennial sea ice. *Geophysical Research Letters*, 34(19).

Notz, D. and Stroeve, J. (2016). Observed arctic sea-ice loss directly follows anthropogenic co2 emission. *Science*, 354(6313):747–750.

Park, D.-S. R., Lee, S., and Feldstein, S. B. (2015). Attribution of the recent winter sea ice decline over the atlantic sector of the arctic ocean. *Journal of Climate*, 28(10):4027–4033.

Popp, M., Schmidt, H., and Marotzke, J. (2016). Transition to a moist greenhouse with co2 and solar forcing. *Nature communications*, 7(1):1–10.

Ritchie, P. D., Clarke, J. J., Cox, P. M., and Huntingford, C. (2021). Overshooting tipping point thresholds in a changing climate. *Nature*, 592(7855):517–523.

Stroeve, J. and Notz, D. (2018). Changing state of arctic sea ice across all seasons. *Environmental Research Letters*, 13(10):103001.

Stroeve, J., Serreze, M., Drobot, S., Gearheard, S., Holland, M., Maslanik, J., Meier, W., and Scambos, T. (2008). Arctic sea ice extent plummets in 2007. *Eos, Transactions American Geophysical Union*, 89(2):13–14.

Strogatz, S. (1994). *Nonlinear dynamics and chaos*. Westview Press.

Tredicce, J. R., Lippi, G. L., Mandel, P., Charasse, B., Chevalier, A., and Picqué, B. (2004). Critical slowing down at a bifurcation. *American Journal of Physics*, 72(6):799–809.

Wagner, T. J. and Eisenman, I. (2015). How climate model complexity influences sea ice stability. *Journal of Climate*, 28(10):3998–4014.